Subject Area:
cellular biology/systems biology

Keywords:
DNA damage, bystander effect, plant bioactive compounds, artificial intelligence

Author for correspondence:
Olusola Clement Idowu
e-mail: sola@hexislab.com

# DNA damage in human skin and the capacities of natural compounds to modulate the bystander signalling

Ewa Markiewicz and Olusola Clement Idowu

Hexis Lab, Newcastle Helix, The Core, Bath Lane, Newcastle upon Tyne NE4 5TF, UK

OCI, 0000-0003-2986-506X

Human skin is a stratified organ frequently exposed to sun-generated ultra-violet radiation (UVR), which is considered one of the major factors responsible for DNA damage. Such damage can be direct, through inter-actions of DNA with UV photons, or indirect, mainly through enhanced production of reactive oxygen species that introduce oxidative changes to the DNA. Oxidative stress and DNA damage also associate with profound changes at the cellular and molecular level involving several cell cycle and signal transduction factors responsible for DNA repair or irreversible changes linked to ageing. Crucially, some of these factors constitute part of the signalling known for the induction of biological changes in non-irradiated, neighbouring cells and defined as the bystander effect. Network interactions with a number of natural compounds, based on their known activity towards these biomarkers in the skin, reveal the capacity to inhibit both the bystander signalling and cell cycle/DNA damage molecules while increasing expression of the anti-oxidant enzymes. Based on this infor-mation, we discuss the likely polypharmacology applications of the natural compounds and next-generation screening technologies in improving the anti-oxidant and DNA repair capacities of the skin.

## 1. Introduction

Skin is the largest organ of the human body, with primary functions in protec-tion against molecular and physiological damage caused by the environment, and profound capacities to undergo processes of regeneration to repair such damage. It is well recognized that deficiencies in the repair mechanisms of the environmental damage, which can be genetically or non-genetically based, are linked to cellular stress and irreversible changes linked to ageing and premature ageing of the skin [1]. One of the major skin-damaging environ-mental factors is ultraviolet radiation (UVR), which is responsible for both the direct introduction of mutagenic changes to DNA and indirect effects executed by reactive oxygen species (ROS) and photosensitizing reactions linked to expression of the redox-sensitive signalling molecules [2]. Such molecules have been consistently documented to be involved in the induction of the bio-logical responses in non-irradiated neighbouring cells, including DNA mutagenesis and alterations in gene expression or cell cycle, as a result of the signals received from irradiated cells and defined as the bystander effect. At present, there is an increasing recognition that the damage to genomic DNA is a complex process linked to both direct and indirect exposure of a cell nucleus to radiation, which inevitably involves rapid as well as delayed biochemical responses to such exposure. The biological relationship between irradiated and non-irradiated cells in the bystander effect, in particular the amplification of the original damaging signal, was originally estimated for α particles, demonstrating that irradiation of 1% of the cells can lead to corresponding DNA damage in more than 30% of the cells [3,4]. Presently, the bystander

royalsocietypublishing.org/journal/rsob Open Biol. 9: 190208

effect is recognized as a significant component of the UVR mode of action, stressing the importance of the search for potential novel molecules with the ability to modulate signalling for enhanced radioprotection [5].

Natural bioactive compounds (NBC), a large group of secondary metabolites from plant sources, have broadly documented protective capacities to counteract the oxidative damage both endogenously in the plant cell and in the animal cell through exogenous applications. These capacities are primarily thought to associate with ROS-scavenging activity determined by specific chemical structures of the compounds. However, the NBC also show complex inter-actions with the DNA damage and oxidative stress biomarkers, affecting their expression or activity in a multi-tude of biochemical assays. They therefore have broader applications in protection against photo-damage [6].

Importantly, the activity of a single NBC is rarely limited towards one specific target; instead, the compounds can act synergistically on multiple molecular targets affecting several biological pathways simultaneously. This suggests that the biological processes relevant to homeostatic balance within multi-cellular tissue, including levels of DNA damage and modulation of bystander signalling, can be significantly enhanced through appropriate selection and validation of var-ious NBC using next-generation screening technologies and a polypharmacological approach. In this manuscript, we discuss the protective applications of NBC in this context, focusing on the skin as an important target in applied biomedical and cosmetic science.

## 2. UVR-induced DNA damage and repair mechanisms in skin cells

Solar UVR is one of the major environmental DNA-damaging factors that lead to photoageing of the skin. UVR affecting the skin belongs to the invisible spectrum of electromagnetic radiation including UVB (280–315 nm) and UVA (315–400 nm), which account for around 95% of all UVR [7]. Both types of radiation penetrate the epidermis; however, only UVA can reach the dermis since it is penetrating the skin more deeply than UVB [8].

### 2.1. Direct and indirect damage caused to DNA by UVR

UVR, particularly UVB, is absorbed directly by nuclear DNA leading to its damage. UV photons induce GC to TA tran-sitions and formation of cyclobutane pyrimidine dimers (CPD) and less abundant but more mutagenic pyrimidine (6–4PP) photoproducts in the DNA [9–11]. The formation of these lesions is influenced by the state of DNA condensation in a cell nucleus; for example, regions sensitive to CPD formation contain telomeres and 5-methyl cytosine in hetero- and euchromatin, whereas 6–4PPs are uniquely formed in the euchromatin [12–14].

Nuclear DNA is also susceptible to indirect damage, which occurs through absorbance of UV photons by the non-DNA chromophores, leading to the formation of ROS and subsequent DNA photoproducts. Both UVA and UVB stimulate the production of ROS; however, oxidative pro-cesses involving mutagenic and cytotoxic effects on DNA are induced mainly by UVA radiation, which has a relatively

poor efficiency in introducing direct DNA damage compared to UVB [15,16].

Human skin contains numerous non-DNA chromo-phores, including riboflavins, porphyrins, haem, bilirubin, melanin precursors, pterins, flavins, tryptophan and urocanic acid [17–19]. The molecules have the capacities to absorb UV photons, which lead to a transient change in molecular struc-ture and the formation of photo-excited state intermediates, with photosensitizing reactions involving energy transfer from triplet state energy electron donors to substrate molecules such as DNA and molecular oxygen [17,20].

Type I reactions give rise to superoxide ($O_2^-$), hydrogen per-oxide ($H_2O_2$) and hydroxyl radical (OH), while type II reactions result in the formation of singlet oxygen ($^1O_2$) that frequently oxidizes guanine in DNA to 8-oxo-7,8-dihydroguanine (8-oxoG). In addition to single-strand breaks (SSB), pyrimidine oxidation products, 8-, 5- or 4-hydroxyadenine and apurinic sites, 8-oxoG is the main oxidized base detected in DNA upon UVR and ROS exposure that can result in a mismatched pairing with adenine leading to G to T and C to A substi-tutions. ROS, particularly hydroxyl radical, also induce the formation of 8-hydroxydeoxyguanosine (8-OHdG), which is one of the most abundant products of oxidative stress [21–24].

Generation of ROS leads to oxidation of other major macromolecules, which has a further destabilizing indirect effect on DNA. Examples include lipid peroxidation, which generates unsaturated aldehydes causing base alkylation in DNA, protein oxidation that blocks DNA replication and transcription or oxidization of melanin by peroxynitrite with subsequent energy transfer to DNA with the formation of CDPs [25–28].

Several molecular mechanisms of DNA damage responses (DDR) present in skin cells counteract the harmful effects of UVR. Unrepaired DNA damage including CDP, 6–4PP and oxidized bases that distort the DNA structure can block the activity of DNA and RNA polymerases, therefore limiting the rate of replication and transcription processes [29,30]. DNA damage normally limits the cellular prolifer-ation through temporary cell cycle arrest that facilitates the initiation of repair mechanisms to prevent propagation of mutagenic lesions [31].

### 2.2. Major DNA repair mechanisms

Two main mechanisms of DNA repair are base excision repair (BER) and nucleotide excision repair (NER). BER is responsible for removing small base lesions such as oxidized 8-oxoG or deaminated adenine and repair of the SSB. BER occurs in sev-eral steps catalysed by major enzymes: AP-endonuclease (APE1), DNA polymerases (Pol $\beta/\delta/\varepsilon$), flap endonuclease (FEN1) and DNA ligases (LIG3/XRCC1 or LIG1) [32,33].

NER is particularly important for repair of DNA damage induced directly by UVR, such as 6,4-photoproducts and thy-mine dimers. NER is initiated by recognition of the damage by replication protein A (RPA) and xeroderma pigmentosum, removal of the single-stranded fragment by transcription factor II H (TFIIH, containing ATP-dependent DNA helicases XPB and XPD) and XPF-ERCC1 and XPG endonucleases, synthesis of the complementary sequence by replication factor C (RFC), PCNA and (Pol $\delta/\varepsilon/\kappa$) and ligation by LIG1/FEN1 or LIG3/XRCC1 to form a double-stranded DNA. NER occurs in two forms—global genomic NER (GG-NER) and transcription-coupled NER (TC-NER)—that

differ in the initial steps of DNA damage recognition. GG-NER repairs damage in both transcriptionally active and silent regions of DNA, using the proteins that recognize structural distortions such as DNA damage binding (DDB, XPE) and complementation group C (XPC-Rad23B). TC-NER repairs damage only in the transcriptionally active regions, which use RNA polymerase (RNApol) stalls for DNA damage recognition [34–36]. The ageing process of human skin is associated with an observed decline in both BER and NER efficiency. Repair efficiency of 8-oxoG and CPD is decreased in the fibroblasts from aged individuals; moreover, 8-oxoG accumulates already in the middle-aged cells, indicative of relatively faster decrease in BER capacity with age [37].

# 3. Association between DNA damage and age-related cell signalling through oxidative stress

## 3.1. Generation of ROS in cellular compartments

In addition to the genomic DNA, the process of photoageing also affects mitochondrial DNA (mtDNA). UVR frequently leads to 4977 base pair deletion of DNA, which in turn contributes to enhanced production of ROS by mitochondria and increased levels of ROS-induced mtDNA damage in a feedback cycle [38].

In the cell, ROS is constantly produced in various organelles as a by-product of aerobic metabolism and human skin remains in direct contact with the atmospheric oxygen.

Electron transport chain (ETC) located in the inner mitochondrial membrane proceeds through complex I (NADH:ubiquinone oxidoreductase), complex II (succinate dehydrogenase), complex III (coenzyme Q: cytochrome $c$ oxidoreductase) and complex IV (cytochrome $c$ oxidase) to $O_2$, resulting in the production of $H_2O$. The electrons can also react prematurely with $O_2$ at complex I and III, leading to the formation of $O_2^-$ instead of $H_2O$ [39]. In parallel, mitochondrial nitric oxide synthase (NOS) produces nitric oxide (NO) that can combine with $O_2^-$ to form peroxynitrite (ONOO−) [40].

Peroxisomal ROS production is driven by flavoenzymes and oxidoreductases, mainly acyl-CoA oxidase (ACO), D-amino acid oxidase (DAO), D-aspartate oxidase (DDO), L-pipecolic acid oxidase (LPIPOX), L-hydroxyacid oxidase (HAO1) and the polyamine oxidase (PAO). The enzymes are involved in the oxidation of fatty acids as well as metabolism of amino acids, glyoxylate and dicarboxylate and produce mainly $H_2O_2$ as a by-product of these reactions [41]. In addition, NOS catalyses the oxidation of L-arginine to NO; in the absence of L-arginine, it leads to the formation of $O_2^-$ [42].

ROS are also produced in the endoplasmic reticulum (ER) through oxidation–reduction reactions involving the cytochrome P450, protein disulfide isomerase (PDI) and oxidoreductin-1 (ERO1). Incomplete transfer of electrons from Ero1 to $O_2$ can result in the formation of $O_2^-$ [43–45].

Finally, ROS can be generated in cytosol as a by-product of the enzymatic activities of cyclooxygenase (COX) and lipoxygenase (LOX) using arachidonic acid as a substrate for synthesis of prostaglandin H2 and leukotrienes. Cellular levels of arachidonic acid are increased in ageing skin, and both COX and LOX have the capacity to produce $O_2^-$ [46,47].

## 3.2. Involvement of ROS in signal transduction and cell proliferation

$H_2O_2$ is recognized as a secondary messenger with the capacity to activate several redox-sensitive signalling molecules involved in the regulation of cell proliferation and migration. The main molecular targets of these responses are mitogen-activated protein kinases (MAPKs), extracellular signal-regulated kinases (ERK1/2), c-Jun NH2-terminal kinases (JNK 1/2/3), phosphoinositide 3-kinase/serine-threonine kinase (PI3 K/Akt), protein kinase B (AKT) and transcription factors such as activator protein 1 (AP-1), nuclear factor κ-light-chain-enhancer of activated B cells (NF-κB) and early growth response 1 (Egr1) [48–52]. The irreversible alterations to these signalling pathways by ROS activate the proto-oncogene pathways that are relevant to skin ageing. Specifically, the delayed generation of ROS by UVR trigger inflammation and cause oxidative stress and DNA damage in non-irradiated neighbouring cells via bystander effect [5,53].

One of the best characterized cellular events that occur in response to damage triggered by UVR exposure is senescence [54–57]. Activation of the senescence programme is associated with irreversible arrest of cell proliferation and development of senescence-associated secretory phenotype [58,59]. Senescent cell phenotype is intrinsically linked to the upregulated and persistent oxidative stress and DNA damage, which have been studied in a number of cell types in the skin, including fibroblasts and melanocytes [60–62]. Generation of the oxidative stress by $H_2O_2$ and $O_2^-$ leads to accumulation of 8-OHdG and the DNA damage is significantly enhanced in the telomeric regions of the chromatin, which are also thought to undergo structural changes in senescence. Senescent fibroblasts also demonstrate an increase in the steady-state levels of 8-OHdG, which can be delayed by treatment with antioxidants [63,64].

Cell proliferation is closely regulated by the signal transduction associated with p53/Rb axis. Tumour suppressor p53 (p53) is a transcription factor normally activated by DNA damage or oxidative stress. Oxidative stress response is mediated by p38 mitogen-activated protein kinases, leading to phosphorylation and stabilization of p53 and transcription of the genes involved in cell cycle arrest. The major regulator of cell cycle arrest mediated by p53 axis is cyclin-dependent kinase inhibitor p21 (p21/Cip1), which controls the activity of cyclin-dependent kinases 2 and 4 (Cdk2/Cdk4) responsible for G1/S-phase cell cycle progression. Activation of p53 is functionally linked to the activation of ataxia telangiectasia mutated (ATM) and ataxia telangiectasia and Rad3-related (ATR) protein kinases via DNA response elements enabling DNA repair during the cell cycle arrest [32,65–67]. Another mechanism of cell cycle arrest controlled by ATM/ATR in response to DNA damage is the degradation of M-phase inducer phosphatase 1 (Cdc25A), which occurs via checkpoint kinases 1 and 2 (Chk1/2) and leads to the inhibition of the cyclin-dependent kinase 1 (Cdk1)–cyclin B1 complex responsible for G2/M cell cycle progression [68,69]. p53 has also the capacity to downregulate expression of Cdc25A, thus establishing additional mechanisms of cell cycle progression in response to stress [70].

Retinoblastoma protein (pRb) is a tumour suppressor responsible for transitions from the G1 to S-phase of cell cycle through interactions with transcription factor E2F. The G1/S-phase transitions are dependent on the phoshorylation

royalsocietypublishing.org/journal/rsob Open Biol. 9: 190208

status of pRb, which is regulated by cyclin-dependent kinases Cdk4 and Cdk6, allowing phosphorylated pRb to release E2F and entry into S-phase. Downregulation of pRb phosphorylation by cyclin-dependent kinase inhibitor p16 (Ink4a) results in the suppression of E2F target genes and cell cycle arrest [71–73]. Expression of p16 is increased by redox-sensitive kinases Erk1/2 and p38 during the oxidative stress and modulation of p16 levels correlates with activation of senescence programme [72,74,75]. Crucially, accumulation of both p16 and p53 is also associated with premature cell senescence, highlighting the central role of p53/Rb axis in control of stress-induced cellular ageing [76].

# 4. Stratification and anti-oxidant defences of the skin

The skin is a stratified organ that has an endocrine function and provides protection against harmful factors in the environment. The skin is composed of two main layers: the epidermis and dermis. The main cells of the epidermis are keratinocytes, which are organized in several defined layers based on the localization, shape, orientation and expression of the biomarkers. The cells undergo successive programmes of divisions, differentiation and migration from the stratum basale to the stratum granulosum and stratum spinosum, leading to the formation of the stratum corneum that contributes to the skin barrier function [77].

Stratum corneum (cornified envelope) demonstrates distinctly high concentrations of low molecular weight (LMW) antioxidants such as vitamin E, vitamin C, ubiquinol, uric acid and glutathione, particularly in the deeper layers of the compartment [78,79]. A main molecule with the anti-oxidative capacity is glutathione (GSH), which reacts directly with ROS such as $H_2O_2$, $O_2^-$ and OH forming intermediate homodimers via disulfidic bond (GSSG). Oxidized thiol is subsequently reduced by glutathione reductase (GR) in the presence of NADPH to restore GSH [80,81]. Concentrations of GSH are additionally reduced in photoaged skin [82].

LMW antioxidants, together with cysteine-rich small proline-rich proteins (SPRRs), counteract the increase in ROS levels and DNA damage in the cornified envelope, which also undergoes dramatic alterations during skin ageing [83].

The epidermis contains a range of ROS-detoxifying enzymes, which are highly concentrated in the stratum granulosum, lowering ROS levels and providing protection against oxidative stress in suprabasal keratinocytes [84,85].

Skin contains several anti-oxidant enzymes that control the intracellular concentrations of $O_2^-$ and $H_2O_2$. Superoxide dismutase (SOD) converts $O_2^-$ to $H_2O_2$ and $O_2$ [86]. Three isoforms are expressed in the skin: SOD1 in cytosol, SOD2 in mitochondria and SOD3 in the extracellular matrix, with UV irradiation leading to the increase in SOD2 expression [87,88]. Deficiencies in SOD1 or SOD2 are additionally associated with skin atrophy and epidermal thinning characteristic of aged skin [89–91].

Catalase (CAT) is localized to peroxisomes where it reduces $H_2O_2$ to $H_2O$ and $O_2$ [92]. CAT is expressed particularly abundantly in the stratum corneum where it shows a decreasing gradient of activity towards the surface of the skin [93,94] Epidermal CAT activity is moreover increased in aged and photoaged skin correlating with increased production of ROS [93,95].

Glutathione peroxidase (GPx) catalyses the reduction of $H_2O_2$ to $H_2O$ in the reaction recycling the GSH and GSSG in the presence of GR/NADPH [96,97]. GPx activity is altered in aged and photoaged skin; with GPx downregulation additionally leading to epidermal hyperplasia [98].

Peroxiredoxins (PRDXs) detoxify ROS through redox reactions involving cysteines forming disulfide bonds at the active centre and subsequent regeneration of the enzyme by thioredoxin reductase (TrxR) [99,100]. PRDX1 and PRDX2 are expressed at high concentration in the stratum granulosum of the epidermis and expression can be additionally induced by UVR [101,102].

Another class of antioxidants is iron-binding transferrin (TF) that blocks formation of OH in Fenton reactions [103].

Compared with the epidermis, the dermis contains relatively low concentrations of both ROS and antioxidants [83]. However, UVA that penetrates the deeper skin compartments can activate the transcription factors controlling the expression of the genes associated with ROS defence in fibroblasts such as nuclear factor erythroid 2-related factor 2 (Nrf2) [104].

Ageing of the skin is associated with further reduction of antioxidants such as GSH in the dermis [82]. Compared with young skin, both dermal concentration of GSSG and GSSG : GSH ratio are decreased in ageing [105]. These changes are accompanied by a decrease in the activity of ROS-detoxifying enzymes; for example, catalase activity is decreased in the dermis of aged and photoaged skin [94,95]. Deficiencies in other anti-oxidant enzymes, SOD2 or GPx are associated with the phenotypes consistent with skin ageing such as atrophy of dermal connective tissue, dermal inflammation and increased levels of Cox2 [89–91,98].

Skin is maintained by direct communication between the dermal and epidermal compartments and the signalling interactions that originate in the dermis are considered to be vital for proper activity of the epidermal progenitor cells [106]. Changes in the epidermis associated with DNA damage and oxidative stress lead to a decrease in cellular turnover rate, terminal differentiation and organization of the stratum corneum, ultimately affecting skin barrier function [107]. In ageing skin, while the epidermal stem cell (EpiSC) population is protected from ROS damage, the transit-amplifying (TA) cells are more abundant, with a longer cell cycle and consequently less differentiation capacity compared to younger counterparts [108–110]. Decreased barrier function could be caused by not only diminished anti-oxidant defences but also signalling from the deteriorating dermal environment affecting TA population in the epidermis.

# 5. The effect of natural compounds on skin cells through network interactions with DNA damage and oxidative stress responders

## 5.1. ROS-regulated signal transduction and radiation-induced bystander signalling

Cellular damage caused by UVR triggers molecular responses not only in the cells directly affected by UV photons but also in the non-irradiated, neighbouring cells via signalling known as the bystander effect [5,53]. The bystander effect involves the molecular signalling interaction between irradiated and non-irradiated cells, and has been

originally identified as a biological phenomenon of low-dose ionizing radiation (i.e. $\alpha$ particles and X-rays); however, it is presently also recognized as part of the cellular response to UVR in biological systems. It is thought that in human skin, a UVR-induced bystander effect would be propagated by inflammatory responses and ROS causing further oxidative damage and genomic instability to DNA in non-exposed proximal cells in the tissue [111].

The bystander mechanism relies on the multiple signalling cascades involving redox and cell cycle signalling molecules coded by COX2, ERK1/2, MAPK, JNK, AP-1 and NFκB genes. Cox2 has been additionally demonstrated as one of the central components of the radiation-induced bystander signalling in human dermal fibroblasts. The bystander response also involves reactive nitrogen species and activation of iNOS. At present, several factors such as inhibitors of gap junction-mediated cellular communication, free radicals scavengers or anti-oxidant enzymes SOD and CAT can decrease the bystander response and damage in the neighbouring cells [112–114].

Currently, there is great interest in the identification and application of undiscovered molecules with the capacity to regulate the DNA damage and bystander signalling. Selective regulation of some of the DNA and oxidative stress markers would allow the protection and repair of the complex tissue by addressing the amplified damage that propagates beyond the individual cell.

## 5.2. Anti-oxidant capacities of natural bioactive compounds

One potentially important and rich source of such activities that is still being characterized is the class of secondary metabolites from plants. When analysed for the effects on the protein targets in skin cells, these NBC have the capacity to downregulate the expression or activity of ROS-activated signal transduction factors and cell cycle/DNA damage proteins while simultaneously upregulating the anti-oxidant enzymes. Examples include compounds with the capacity to inhibit redox-sensitive signalling, which belong to a group of flavonoids ((+)-catechin), monoterpenes (eucalyptol, sabinene) and polyphenols (mangiferin, sauchinone). These compounds downregulate the expression or activity of JNK, p38, Cox2, iNOS, AP-1 and Erk1/2 [115–119]. Compounds belonging to a group of phenolic diterpenes (carnosic acid, carnosol), catechins (epigallocatechin-3-gallate) or phenolic acids (ferulic acid, caffeic acid) also increase the activity of ROS-responsive transcription factors such as Nrf2 or ROS-detoxifying enzymes such as CAT and GPx [120–122]. Both groups of skin cell biomarkers are also responsive to isoflavones (acetyldaidzin, acetylgenistin, acetylglycitin, daizein, 5,7-dimethoxyflavone, genistein, glycitein) and phenolic acids (caffeic acid), which inhibit Cox2, Erk1/2, JNK, p38, iNOS, MAPK and NFκB while activating CAT or GPx [123–127]. Finally, oxidative stress markers can be downregulated concomitantly with the cell cycle/DNA damage proteins such as p53 or p21 by the compounds belonging to diterpenes (acanthoic acid), flavonoids (baicalin) and polyphenols (butein, curcumin, sativanone, sesamin) [128–132].

Secondary metabolites with the capacity to modulate redox signalling and protect against oxidative damage are abundantly found in plant tissue where they play an important role in the defence system and tolerance to a range of environmental stresses that can lead to excessive production of ROS [133–135]. The anti-oxidant properties of the compounds are generally determined by their chemical structure, based on aromatic rings with one or more hydroxyl (–OH) or methoxy (O-CH$_3$) group [136]. Phenolic compounds use these structures by donating electrons or hydrogen atoms and chelating metal cations such as Fe$^{2+}$ to neutralize and inhibit formation of free radicals [137,138]. Flavonoids constitute a group of phenolic compounds with three rings (A/B phenyl benzopyrone and C pyran) as a basic structure and low redox potential of hydroxyl group. Flavonoids act as scavengers of free radicals, forming oxidized forms that are more stable and less reactive while reducing superoxide, peroxynitrite and hydroxyl radicals through hydrogen donations [137–140].

The NBC could also have the potential to regulate a network of activities and repair mechanisms that are triggered by UVR-induced direct DNA damage and secondary redox-related cellular responses in the skin environment. In addition to direct effects on ROS, the NBC demonstrate likely capacities to affect skin cells in three major areas: (i) inhibition of the proteins involved in cell cycle arrest and senescence, enhancing DNA repair mechanisms induced by DNA photoproducts and DNA oxidation; (ii) activation of ROS-detoxifying enzymes and transcription factors, enhancing the natural anti-oxidant defences in the epidermis and counteracting the cellular ROS as part of by-products of metabolism and UVA damage; and (iii) inhibition of redox-sensitive signalling molecules, activated primarily in the dermis by UVA-induced ROS and constituting a part of the bystander signalling. Collectively, the NBC would fulfil a protective role against propagation of the oxidative stress and DNA damage, amplified stress response and premature ageing of the tissue (figure 1).

## 5.3. Polypharmacology characteristic of NBC versus amplified damage in bystander signalling

Presently, the biological effects of radiation, in addition to DNA damage upon direct cellular exposure, are also recognized to include the non-DNA-targeted effects associated with the bystander effect, adaptive response and radiosensitivity, which are particularly evident at low doses [141–145]. The cellular damage in this instance can be recorded to include the aberrations in DNA or changes in cell cycle and gene expression observed in unirradiated cells as a result of interactions with irradiated cells [146–149]. In classical evidence, a significantly higher amount of DNA damage is detected than it would be expected based on calculations of cell nuclei directly targeted by the radiation, indicative of signal transduction pathways and extracellular factors responsible for the bystander effect [4,150,151]. Similarly, higher fractions of cells with altered gene expression and a significant rise of damaged cells to those normally expected from directly exposed cells are associated with the bystander effect, including human fibroblasts showing a two to threefold increase in the level of damage [152,153]. The radiation-induced bystander effect or tissue responses to low-dose radiation have been associated with occasional stimulation of pro-mitogenic activities *in vitro*, linked to expression of p53 and p21 or cyclin D1

royalsocietypublishing.org/journal/rsob    Open Biol. 9: 190208

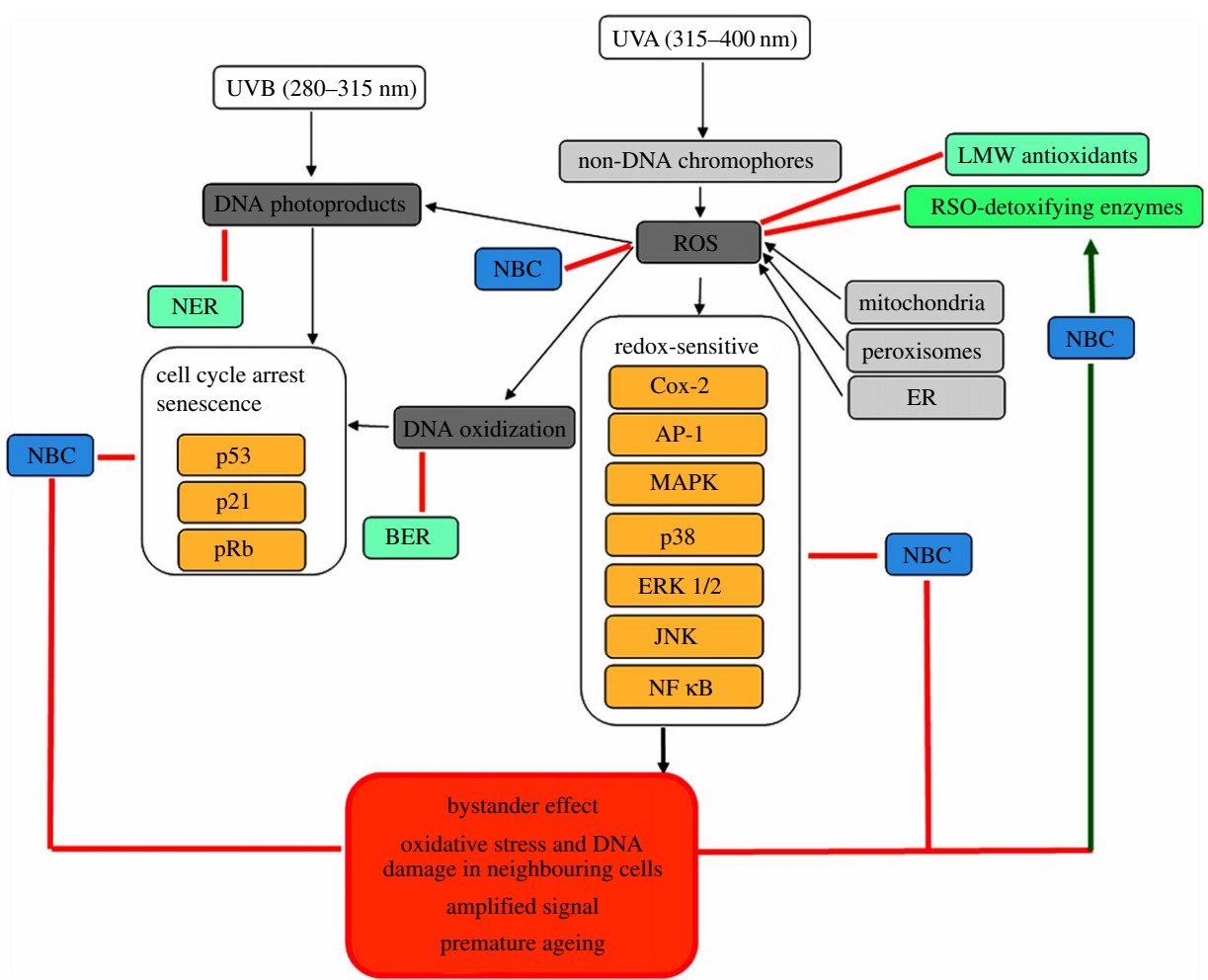

**Figure 1.** Schematic of the proposed capacities of the NBC in a context of protection against UVR-induced DNA damage in the skin. UVB and UVA affect the DNA either directly to induce the range of photoproducts or indirectly, through non-DNA chromophores and ROS, leading to DNA oxidization. Both forms of DNA damage trigger the repair mechanisms (BER and NER). Persistent DNA damage associates with cell cycle arrest and can lead to cellular senescence, which is executed primarily by p53/pRb axis. Synthesis of ROS also occurs as part of the metabolic reactions in various cellular organelles and is counteracted by LMW antioxidants and ROS-detoxifying enzymes, which are particularly abundant in the epidermis. Redox-sensitive proteins are activated by the oxidative stress caused by UVA, presumably predominantly in the dermis, also forming a part of the signalling causing amplified damage in the neighbouring cells known as the bystander effect. NBC have the capacity to affect the cell cycle arrest/DNA repair axis, stimulate the ROS detoxification and inhibit the redox-sensitive signalling, leading effectively to the inhibition of amplified damage and skin protection.

[151,154]. In addition to activation of biological effects within a low-dose range, the bystander effect also shows a nonlinear dose dependence, with the shape of a dose-dependent curve demonstrating a flattening rather than statistically significant increase with increasing dose, which reflects a saturation of the response above the threshold level [4,155]. The bystander effect therefore amplifies the original damage present within a fraction of the cellular population, which is executed by the redox-dependent signalling pathways as one of the possible outcomes.

The polypharmacology nature of the NBC, namely the ability to affect multiple protein targets simultaneously, could indicate the intrinsic capacities of the molecules to modulate the bystander signalling by synergistic cascade reactions serving as a protective mechanism against tissue damage. NBC working in an integrated system composed of multi-cellular microenvironment and multi-target protein factors could facilitate better recognition of the damage and repair or alternatively replacement of the permanently damaged cells with new progeny or induction of differentiation. It would remain to be identified if the cells highly susceptible to NBC belong to radiosensitive or radioresistant populations. Based on this information, a working model could be proposed to demonstrate the potential biological responses in the multi-cellular tissue exposed to irradiation and NBC. The damaged cells would produce bystander signals, which would be received by and processed within a sensitive subpopulation of the cells, leading to alterations in cell cycle, differentiation or senescence. The presence of NBC would have a radioprotective effect through interactions with the cascade of signal transduction molecules involved in amplification of the bystander signals, potentially enhancing the mechanisms contributing to the replacement of damaged cells and DNA repair (figure 2).

Molecular effects of various NBC can be captured through *in silico* interactive networks, demonstrating the synergistic capacities to affect the expression or activity of multiple cellular factors across many pathways. This indicates that the biological activity of the molecules extends far beyond their role as ROS scavengers. Networks involving compound–protein interactions additionally reveal the capacities of NBC as multi-target ingredients, which can also identify the enriched or more conserved clusters involved in such interactions. For example, isoflavones such

royalsocietypublishing.org/journal/rsob    *Open Biol.* **9**: 190208

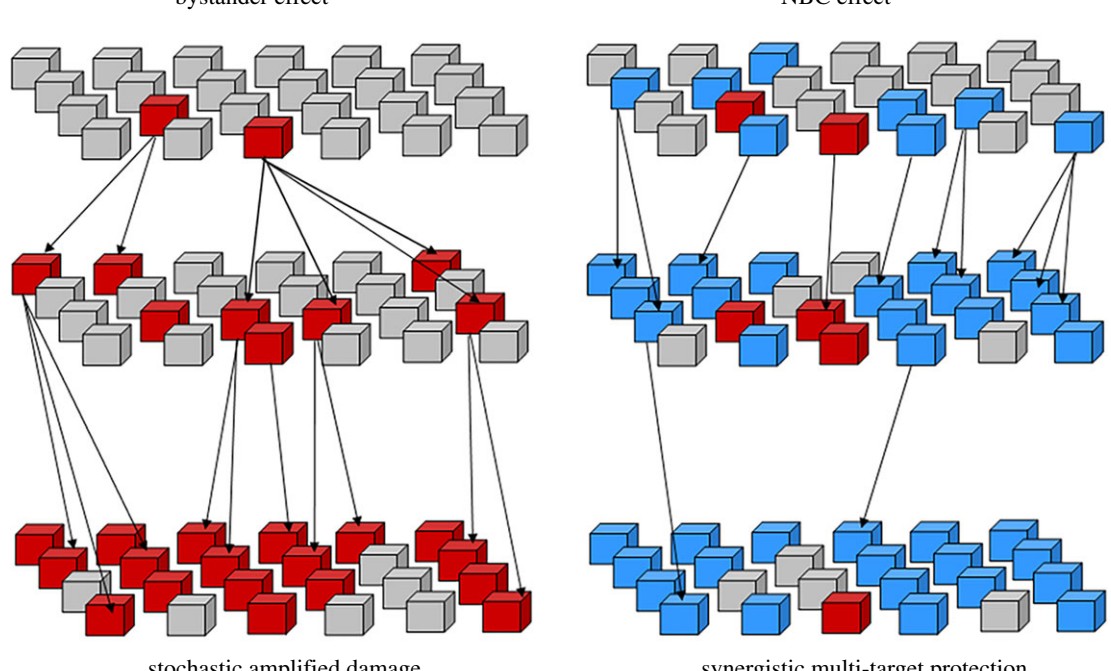

**Figure 2.** Schematic diagram illustrating proposed interactions between bystander signalling and NBC through redox- and DNA damage-sensitive molecules. The bystander effect commences with low-level damage induced by interaction of the cell nucleus with radiation. The changes are subsequently amplified in a stochastic manner, leading to an increased population of the cells with alterations in DNA, gene expression and cell cycle kinetics (red squares). NBC (blue squares) have the capacity to downregulate the expression or activity of redox-sensitive signal transduction factors and cell cycle/DNA damage proteins while simultaneously upregulating the anti-oxidant enzymes. This can be executed in synergy, involving multiple protein targets simultaneously, therefore also amplifying the NBC-protective signals. These characteristics would afford the NBC likely capacities towards enhanced modulation or inhibition of the bystander effect while stimulating DNA repair and regeneration of damaged cells.

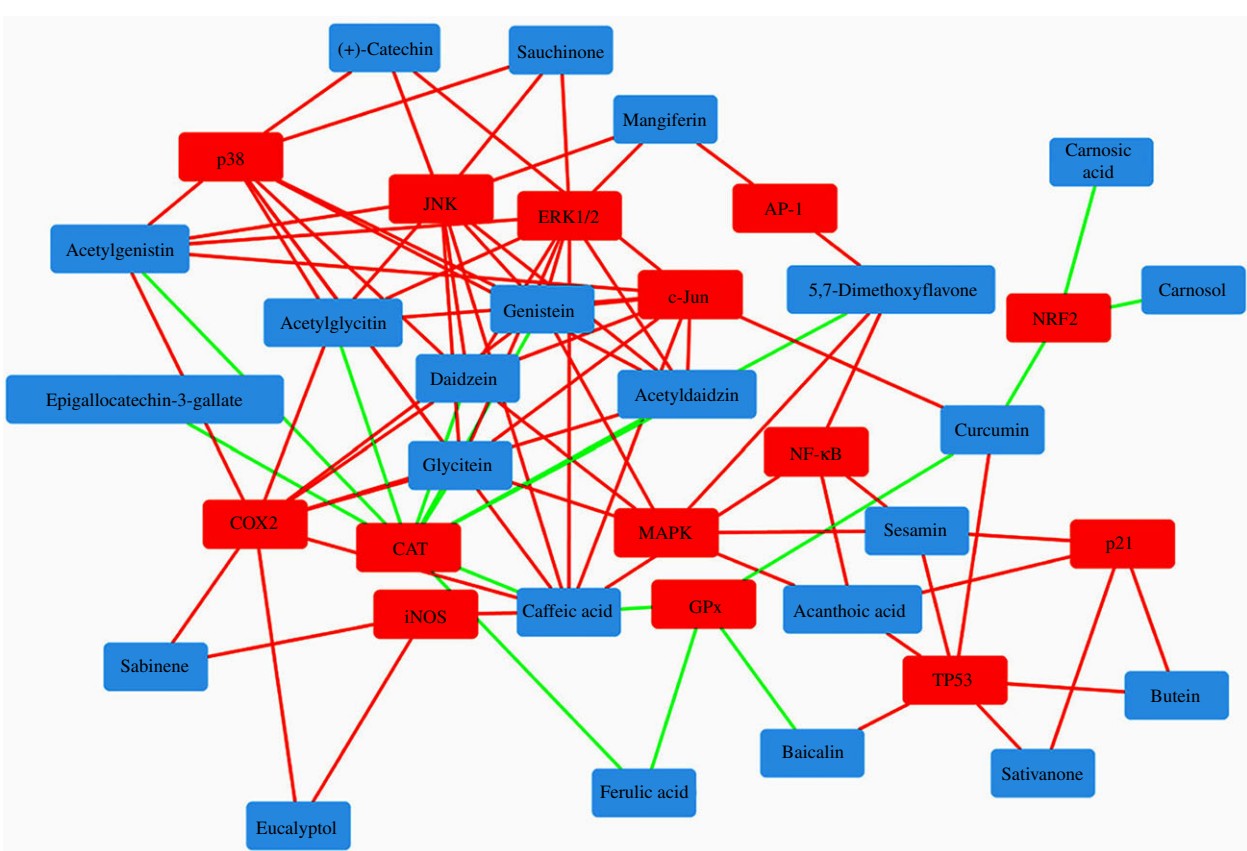

**Figure 3.** Interactive networks created using HexisLab Pro.X *in silico* platform showing functional links between NBC (blue) and skin cellular targets involved in DDR and redox homeostasis (red). The *in silico* network is built based on the manually curated data from the literature [115–132]. Red links represent downregulated expression/activity; green links represent upregulated expression/activity.

as daidzein, genistein and glycitein could fulfil a candidate role as modulators of multiple redox-signalling interactions while cell cycle processes could be enhanced by synergistic activities of sativanone and butein. This illustrates that the NBC can act synergistically on multiple molecular targets affecting several biological processes simultaneously, consistent with the mechanisms underlying a polypharmacology approach (figure 3). The synergistic and multi-targeted capacity of NBC can be particularly applicable in potential interactions able to counteract or modulate the molecular network involved in bystander signalling, which remains as yet unresolved.

# 6. Conclusion and future perspectives

Exposure to environmental UVR is one of the major factors linked to DNA damage in the skin cell. In addition to direct damage, UVR is also a potent inducer of ROS that cause oxidative changes to DNA and activation of redox-responsive signalling molecules and pathways. Some of these factors have been identified as part of the signalling known as the bystander effect, which induces changes and amplifies the original damage present in irradiated cells. Natural compounds from plant sources have the capacity to inhibit the expression or activity of these factors while simultaneously improving the anti-oxidant and DNA repair capacities of skin cells. Recent steady advances in modern skincare formulations also require inclusion of UVR-protective factors. Inhibition and repair of the direct and indirect UVR damage is relevant to a number of skin conditions, including sunburn, uneven skin tone, ageing or photo-sensitive atopic dermatitis. Many of the biomarkers involved in

UVR response would be relevant to other skin conditions caused by environmental exposure such as chemical air pollution.

Application of natural, chemically stable and safe ingredients in the dermatology and cosmetic industry represents a promising avenue and a growing trend in the design of novel treatments. The list of compounds discussed here is not comprehensive; however, it can be expanded and appropriately verified using next-generation platform discoveries such as artificial intelligence (AI), deep learning algorithms and *in silico* screening of biomolecular libraries from natural sources. Such technologies are presently established; in addition, they also enable a fast and cost-effective design and validation of new bio-based products. The technologies resolve the high complexity of biological systems and enable the review of alternate functions of the molecules to repurpose them for new applications, for example, radioprotection. Such an approach will allow comprehensive rapid selection and validation of a range of bioactive compounds with predicted capacities to protect against or repair the molecular damage either individually or in synergy and enhance the desired effect on the skin dictated by future biomedical and cosmetic applications. Future studies based on the natural compounds could also provide more insight into the nature of bystander signalling, to enable its modulation relevant to therapeutic and cosmetic purposes in the field.

Data accessibility. This article does not contain any additional data.
Authors' contributions. Both authors contributed to the manuscript and figures production.
Competing interests. We declare we have no competing interests.
Funding. We received no funding for this study.

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
