## [Reviewer comments · Open Biology]

Review History

RSOB-19-0208.R0 (Original submission)

Review form: Reviewer 1

Recommendation

Accept with minor revision (please list in comments)

Do you have any ethical concerns with this paper?

Yes

Comments to the Author

This review article addresses a potentially important area of skin biology that deals with understanding the biological endpoints of UV-induced DNA damage and how such endpoints can be modified by NBCs. The authors have done a good job in organizing the review into main sections representing the key areas in the responses to DNA damage in skin cells and the current evidence that NBCs can ameliorate many of the deleterious effects of UV exposure. The authors' highlighting the tools used by their company to identify interaction networks (Fig. 3) could be construed as promoting a commercial venture within the context of a scholarly review. It is

suggested that direct references to the company be replaced by neutral statements with appropriate references.

Specific Comments

1. Section 2 heading is misleading as it reviews mostly cellular studies in general and not in the skin.
 2. Page 5 lines 110, 111. I believe the authors mean "deaminated adenine" and not "deaminated hypoxanthine."
 3. Page 6. The level of detail presented for the discussion of BER is excessive - this is not a DNA repair review.
 4. Page 5. Statement about TC-NER is incorrect. This process repairs repairs DNA damage on template strands of transcribed genes but not on the non-template strands of those genes.
 5. Title of section 5 is awkward - suggest replacing "biomarkers" with "responders."
 6. Page 16, line 353. insert "damage" after "DNA."
- Page 19, lines 414-416. Eliminate sentence highlighting the authors' company-developed technologies and replace with web reference.

The manuscript should be thoroughly edited by an English-proficient editor and the numerous grammatical and usage errors corrected.

Decision letter (RSOB-19-0208.R0)

08-Nov-2019

Dear Dr Idowu,

We are pleased to inform you that your manuscript RSOB-19-0208 entitled "DNA Damage in Human Skin and the Capacities of Natural Compounds to Modulate the Bystander Signalling" has been accepted by the Editor for publication in Open Biology. The reviewer has recommended publication, but also suggest some minor revisions to your manuscript. Therefore, we invite you to respond to the comments and revise your manuscript.

Please submit the revised version of your manuscript within 7 days. If you do not think you will be able to meet this date please let us know immediately and we can extend this deadline for you.

- 1) A text file of the manuscript (doc, txt, rtf or tex), including the references, tables (including captions) and figure captions. Please remove any tracked changes from the text before submission. PDF files are not an accepted format for the "Main Document".
- 2) A separate electronic file of each figure (tiff, EPS or print-quality PDF preferred). The format should be produced directly from original creation package, or original software format. Please note that PowerPoint files are not accepted.
- 3) Electronic supplementary material: this should be contained in a separate file from the main text and meet our ESM criteria (see <http://royalsocietypublishing.org/instructions-authors#question5>). All supplementary materials accompanying an accepted article will be treated as in their final form. They will be published alongside the paper on the journal website and posted on the online figshare repository. Files on figshare will be made available approximately one week before the accompanying article so that the supplementary material can be attributed a unique DOI.

Online supplementary material will also carry the title and description provided during submission, so please ensure these are accurate and informative. Note that the Royal Society will not edit or typeset supplementary material and it will be hosted as provided. Please ensure that the supplementary material includes the paper details (authors, title, journal name, article DOI). Your article DOI will be 10.1098/rsob.2016[*last 4 digits of e.g. 10.1098/rsob.20160049*].

- 4) A media summary: a short non-technical summary (up to 100 words) of the key findings/importance of your manuscript. Please try to write in simple English, avoid jargon, explain the importance of the topic, outline the main implications and describe why this topic is newsworthy.

Images

Data-Sharing

It is a condition of publication that data supporting your paper are made available. Data should be made available either in the electronic supplementary material or through an appropriate repository. Details of how to access data should be included in your paper. Please see <http://royalsocietypublishing.org/site/authors/policy.xhtml#question6> for more details.

Data accessibility section

Sincerely,
The Open Biology Team
mailto:openbiology@royalsociety.org

Reviewer(s)' Comments to Author:

Referee:

Comments to the Author(s)

This review article addresses a potentially important area of skin biology that deals with understanding the biological endpoints of UV-induced DNA damage and how such endpoints can be modified by NBCs. The authors have done a good job in organizing the review into main sections representing the key areas in the responses to DNA damage in skin cells and the current evidence that NBCs can ameliorate many of the deleterious effects of UV exposure. The authors' highlighting the tools used by their company to identify interaction networks (Fig. 3) could be construed as promoting a commercial venture within the context of a scholarly review. It is suggested that direct references to the company be replaced by neutral statements with appropriate references.

Specific Comments

1. Section 2 heading is misleading as it reviews mostly cellular studies in general and not in the skin.
 2. Page 5 lines 110, 111. I believe the authors mean "deaminated adenine" and not "deaminated hypoxanthine."
 3. Page 6. The level of detail presented for the discussion of BER is excessive - this is not a DNA repair review.
 4. Page 5. Statement about TC-NER is incorrect. This process repairs repairs DNA damage on template strands of transcribed genes but not on the non-template strands of those genes.
 5. Title of section 5 is awkward - suggest replacing "biomarkers" with "responders."
 6. Page 16, line 353. insert "damage" after "DNA."
- Page 19, lines 414-416. Eliminate sentence highlighting the authors' company-developed technologies and replace with web reference.

The manuscript should be thoroughly edited by an English-proficient editor and the numerous grammatical and usage errors corrected.

Author's Response to Decision Letter for (RSOB-19-0208.R0)

See Appendix A.

Decision letter (RSOB-19-0208.R1)

19-Nov-2019

Dear Dr Idowu,

We are pleased to inform you that your manuscript entitled "DNA Damage in Human Skin and the Capacities of Natural Compounds to Modulate the Bystander Signalling" has been accepted by the Editor for publication in Open Biology.

Article processing charge

Please note that the article processing charge is immediately payable. A separate email will be sent out shortly to confirm the charge due. The preferred payment method is by credit card; however, other payment options are available.

Sincerely,

The Open Biology Team
mailto: openbiology@royalsociety.org

Appendix A

Dr Olusola Idowu
CEO
Hexis Lab
Newcastle Helix
Newcastle upon Tyne
United Kingdom

Prof. David Glover
Editor-in-Chief
Open Biology

14th November 2019

Dear Professor Glover,

We would like to bring to your attention our response to peer-review evaluation of the article titled 'DNA damage in human skin and the capacities of natural compounds to modulate the bystander signalling' by Ewa Markiewicz and Olusola Idowu, submitted to the Open Biology.

We would like to thank the Reviewer for helpful comments and constructive insights, particularly towards organization and presentation of the scientific knowledge, which in our view helped towards improvement of this manuscript.

Please find below our point-by-point response to Reviewer's evaluations.

Reviewer's comment

1. Section 2 heading is misleading as it reviews mostly cellular studies in general and not in the skin.

Our response: Section 2 heading has now been amended to "UVR-induced DNA damage and repair mechanisms in skin cells" to reflect the information cited as originating from the cellular studies.

Reviewer's comment

2. Page 5 lines 110, 111. I believe the authors mean "deaminated adenine" and not "deaminated hypoxanthine."

Our response: "deaminated hypoxanthine" is now replaced with correct "deaminated adenine"

Reviewer's comment

3. Page 6. The level of detail presented for the discussion of BER is excessive - this is not a DNA repair review.

Our response: The section discussing BER has now been shortened to only two sentences; to reflect the presence of BER mechanism in the cell and the key enzymes.

Reviewer's comment

4. Page 5. Statement about TC-NER is incorrect. This process repairs DNA damage on template strands of transcribed genes but not on the non-template strands of those genes.

Our response: The manuscript now contains the senescence stating “TC-NER repairs damage only in the transcriptionally active regions, which utilise RNA polymerase (RNAPol) stalls for DNA damage recognition” (lines 124-125).

Reviewer’s comment

5. Title of section 5 is awkward - suggest replacing "biomarkers" with "responders."

Our response: The “biomarkers” are now replaced with sounding better in this context “responders”.

Reviewer’s comment

6. Page 16, line 353. insert "damage" after "DNA."

Our response: We inserted “damage” after “DNA” in this sentence (now line 344).

Reviewer’s comment

Page 19, lines 414-416. Eliminate sentence highlighting the authors' company-developed technologies and replace with web reference.

Our response:

Direct reference to the company-developed technologies is now omitted (now lines 404-406)

Reviewer’s comment

The manuscript should be thoroughly edited by an English-proficient editor and the numerous grammatical and usage errors corrected.

Our response:

Entire manuscript has now been systematically checked and corrected by UK English native speaker with an extensive experience in proof reading.

Thank you for kind consideration of the manuscript.

Sincerely,

Ewa Markiewicz and Olusola Idowu